# The Attitudes Towards the Use of Restraint and Restrictive Intervention Amongst Healthcare Staff on Acute Medical and Frailty Wards—A Brief Literature Review

**DOI:** 10.3390/geriatrics4030050

**Published:** 2019-09-04

**Authors:** Ramith Gunawardena, David G. Smithard

**Affiliations:** 1King’s College Medical School, Strand, London WC2R 2LS, UK; 2Consultant Physician, Queen Elizabeth Hospital, Lewisham and Greenwich Trust, Woolwich, Greater London SE18 4 QH, UK

**Keywords:** restraint, direct, indirect, staff, staffing levels

## Abstract

Restraint in modern non-psychiatric-based healthcare is often regarded as a rare occurrence. It is deemed to be used as a last resort to prevent patients from directly harming themselves. However, techniques are used in modern day practice which are considered direct and indirect restraints with the justification of maintaining patient safety, but they are often not classified as “restraints”. Examples of these include the use of bed rails or tables to prevent patients from “wandering” and to reduce the risk of falls and injuries. More indirect techniques would involve passive interactions with patients or leaving mobility aids out of reach. Staff subconsciously restrain patients and reduce their liberties despite agreeing that patient autonomy should be upheld—a necessary evil to maintain a duty of care. Whilst the use of restraints is often justified to ensure patient care and prevent injury, it is not without consequence. There are physical and psychological health risks such as pressure sores from the inability to mobilise, or the brewing of anger and frustration when denied access to everyday actions. The reasons why restraints are used, whilst stemming from maintaining patient safety, are often due to low staffing levels and the inability to constantly watch at-risk patients due to a large workload. Inadequate training is another factor; by improving education in direct and indirect restraint and providing alternative methods, more ethical decisions and positive outcomes can be implemented. Healthcare professionals are reluctant to use restraint but often conduct it without realising it; assessing their understanding of restraint and providing education to raise awareness of the consequences of direct and indirect methods would result in positive steps toward reducing their use at the same time as looking to provide alternatives to uphold patient care whilst maintaining their dignity and liberty.

## 1. Introduction

The use of restraints and restrictive intervention in healthcare is and has been common practice in the UK and internationally [1]. In present times, physical and chemical restraints continue to be used, but this fact is not widely known/obvious. Whilst distinguishable, physical restraints such as wrist restraints are rarely used unless in exceptional circumstances with sufficient justification [2], whereas less obvious forms are used on a daily basis. Examples include the use of bed rails in an attempt to keep patients within the confines of their bed space [3] or the use of an allocated healthcare assistant to make sure that the patient does not cause harm to themselves or others whilst suffering from delirium or a longstanding cognitive impairment [otherwise known as healthcare assistant specials or sitters] [4]. Therefore, there are many different types of subconscious restraints used by clinical staff, but these are often not considered to amount to “restraining the patient”. 

The Care Quality Commission (CQC) (United Kingdom Department of Health) state that all restrictive interventions should be used for the shortest time possible using the least restrictive means to meet the immediate need [2]. Furthermore, the National Institute for Health and Care Excellence (NICE—United Kingdom) recommends the promotion of freedom of movement and the minimization of the use of restraint in patients living with dementia and delirium [5]. However, in practice, the use of restraints is frequently used proactively to prevent potential identified events such as patient falls or abscondment. It is therefore important to consider the attitudes of healthcare professionals towards the use of restraint and determine whether the CQC and NICE guidelines are being followed. 

This literature review will explore the different types of physical restraints used in a non-mental healthcare setting in various countries, as well as when they are used, the reasons why they are used and the attitudes towards their use. There is a vast amount of research into the use of restraints in mental healthcare, especially with the movement towards banning the use of facedown restraints by the Millfield charter and government announcements moving towards the ban of this technique [6,7,8,9]. The knowledge of types of conscious and subconscious restraints will be investigated, as well as how this could be improved so that it can be kept a minimum as per the CQC and NICE guidelines. By looking at the causes of the use of restraint—whether delirium, aggression or short staffing levels—other techniques and methods could be used to reduce the restriction of liberty. Furthermore, by assessing staff attitudes towards restraint, the reasons why restraints are deemed to be necessary could be determined, as well as understanding when alternatives could be used—improving both the patient’s wellbeing as well as mitigating any guilt members of staff would feel. 

## 2. What is Restraint?

Restraint is defined as the “deprivation or restriction of liberty or freedom of action or movement” and may be implemented in numerous ways [4]. Bleijlevens et al. (2016) provide a consensus definition: “Physical restraint is defined as any action or procedure that prevents a person’s free body movement to a position of choice and/or normal access to his/her body by the use of any method, attached or adjacent to a person’s body that he/she cannot control or remove easily [5]”. Similarly, the Centers for Medicare and Medicaid have defined physical restraint as “any manual method, physical or mechanical device, material or equipment attached or adjacent to the patient’s body that he or she cannot remove and restricts freedom of movement or normal access to one’s body” [6].

In the UK, there are two similar legal definitions for restraint in general: the Mental Capacity Act (2005) defines restraint as when someone “uses or threatens to use force to secure the doing of an act which a person resists or restricts a person’s liberty whether or not they are resisting [7]”. Subsequently, in 2019, the Equalities and Human Rights Commission (ECHR) published a framework “to protect and serve to protect and respect the safety and dignity of people being restrained…” to aid compliance with sections 3, 8 and 14 of the ECHR [8].

Increasingly, the use of physical restraint is considered not only inappropriate, but at times demeaning, dangerous and ineffective [10]. However, despite the recognition of the inappropriate use of restraints, they continue to be frequently used in many clinical areas of acute hospitals [11].

## 3. Methodology

An advanced search was used on Pubmed.gov [12]. Initially, the search terms used were “(healthcare) AND restraint” in order to find a broad spectrum of papers. This was then narrowed down to “((healthcare) AND restraint) NOT psychiatry” to find non-psychiatry-based papers. To find papers based on the topics that were covered, “(((healthcare) AND restraint) AND knowledge) NOT psychiatry” as well as “(((healthcare) AND restraint) AND attitudes) NOT psychiatry” were used. Often, sources were found following a search for papers by hand which were yielded from the previous search. Additional papers were sought from Google Scholar. This yielded 44 applicable articles which are discussed in this paper.

As part of this paper, an observational study was carried out at Queen Elizabeth Hospital (Woolwich, UK) to determine the frequency of restraint usage on an Acute Medical and Frailty Ward (a total of 78 beds). On five different afternoons, instances of direct and indirect restraint were tallied and graphed (Table 1, Figure 1 and Figure 2). 

## 4. Frequency of Restraint Use

The use of physical restraint in the care setting, including hospitals, has become standard due to custom and practice and not as a result of an evidence base [13]. There is variability in the use of physical restraint (0–25.5%) between countries and between institutions within those countries. The frequency of restraint use in the United States is 17% in acute settings [13], and 25.5% in Japanese long-stay wards [14]. Older people, those dependent on care and those cognitively impaired were more likely to be restrained [13,14,15]. Minnick surveyed the wards in an acute care hospital over 18 randomly selected days and found that in 50 out of 1000 patient days, restraint was used [16].

The observational study performed at Queen Elizabeth Hospital found frequent use of restraint in the Acute Medical and Frailty Wards 1 & 2 (Figure 1 and Figure 2). The frequency of restraint use varied between 27 and 44. 

Figure 2 demonstrates the frequency of each type of restraint. The bed rails were the most common form of restraint used, confirming results in earlier publications [3]. Each type of restraint was more frequently used on the frailty ward, which may reflect the type of patient rather than a difference in practice. The error bars display the standard deviation, which are overlapping, showing that, whilst a trend is visible, more data are required.

## 5. Types of Restraints

The types of restraints used with in a hospital setting can be separated into two main categories: direct and indirect restraints [17,18]. Direct restraint (Table 1 and Figure 2) refers to equipment being used, whether specialised or not; examples include movable tables, bed sides, restraint belts, bed linen, locking the door or placing the patient into a position in which they cannot get up without assistance from staff. In some cases, nursing staff also used their own physical strength to keep the patient still [18]. Indirect examples include the promotion of passivity, whereby nurses restrict the patient’s movements to only situations pertinent to their care by keeping mobility aids out of reach or employing a healthcare assistant only to make sure that the patient remains seated or does not walk too far from their bed space [19]. Many of those being physically restrained were also prescribed psychoactive medication [15].

## 6. Harmful Consequences from the Use of Restraints

The use of restraints is not without identifiable psychological (to both staff and patients) or physical health risks. As listed in Table 2, physical examples include injury and aggression, pressure sores and loss of muscle tone, contractures, asphyxiation and death due to strangulation [20]. Psychosocial effects can include anger, frustration, aggression, fear, reduced engagement and apathy [20]. Using such methods can also raise ethical issues with staff members and grow feelings of contentiousness and unhappiness [21].

Agitation is common in delirium and psychiatric illness; medical wards may find this difficult to manage and as a consequence resort to physical restraint, which, conversely may result in an increased agitated state [13]. Elevated bedsides could lead to patients trying to climb out of bed and over the bedside with a greater distance to fall, or trying to manoeuvre around obstacles set in place to hinder the patient’s movements [18].

Physical restraint results in the inability to move from the place of restraint and is frequently used for <10 h [15] but may be used repeatedly for many days [14]. Any use—let alone sustained use—can result in full skin thickness damage, physical injury or death [13,15]. The use of a chair or bed restraint can result in the development of pressure ulcers [22], and struggling whilst restrained can lead to friction or shearing/tearing of the skin [22]. The presence of urinary or faecal incontinence [23] secondary to the forced immobilisation will increase the risk of pressure sore occurrence and secondary infection [24,25,26]. Lofgren et al. (1989) found that patients restrained for more than four days have a greater incidence of pressure ulcers, hospital-acquired infections and faecal and urinary incontinence than those restrained for four days or less. 

Brachial plexus or radial nerve injury secondary to restraint is not uncommon; it can result from the combination of vest restraint and the bed positioning causing a patient to slide down the bed and the vest itself putting pressure onto the distal brachial plexus [27]. Tight wrist, arm and ankle restraints will lead to local compression injury and oedema and cyanosis distal to the restraint after prolonged use. Tight chest restraints may prevent ventilation, resulting in asphyxiation and subsequent death [28]. Despite recommendations that the use of restraint techniques should be used at a minimum level in the United Kingdom [2], their use is still reported nationally and worldwide. 

## 7. Reasons for the Use of Restraints

There are many reasons why staff choose to use physical (or chemical) restraint (Table 3), all of which fall under the umbrella of patient safety [29]. These include the care of agitated or restless patients, a lack of staff to adequately watch such patients or to protect patients from injury secondary to wandering which could lead to falls [30,31,32]. In many acute care settings, at-risk patients are often collated to reduce the amount of staff required to monitor them. One-to-one care may be provided if deemed necessary, and if aggression arises, hospital security is called [4].

The prevention of wandering (and subsequently potential falls) and the control of behaviours such as aggression and restlessness [33] are some of the main reasons for the use of restraints. Falls, head injury, femur fracture and subsequent litigation cost the NHS £4.6 million per day [34]. Patients suffering from delirium, dementia or agnosia may not understand the need for a device to be attached to them. The removal of cardiac ambulatory recorders is not life-threatening but pulling at a nasogastric tube could result in feed entering the lungs. It is common to use a nasal bridle to keep a nasogastric tube in place, and such devices, amongst others, are one of the common (48.6%) reasons for the use of restraint in a critical care setting [32,35]. 

## 8. Attitudes from Healthcare Professionals Towards the Use of Restraints

The attitudes towards restraint can vary between healthcare professionals and the public for various reasons. In a survey conducted at a general hospital in Dorset, England, there was 99% agreement between the two groups that fall prevention was important [36]. Furthermore, there was 84.5% agreement that restraint is justifiable to prevent harm. However, there was an 81.5% disagreement that restraint should not be used if the patient is at risk of falling. Out of 100 subjects for each group, 82% of healthcare professionals thought that restraint should always be an option for patients at risk of falling compared to only 45% of patients/relatives. It was also found that 91% of healthcare professionals believe that restraining methods are acceptable at their discretion, compared to 43% of patients/relatives. Patients and relatives were also much more perceptive to the use of furniture rearrangement and bedrails as restraint than healthcare professionals. This highlights a difference between societal and cultural difference in attitudes towards the use of restraints.

## 9. Attitudes Towards the Reasons Why Restraints are Used

A common justification for the use of restraints by healthcare staff as an intervention is to reduce risk factors that may compromise patient safety with altered mental status [37]. One nurse from this study remarked, “Physical restraints that we use here in the surgical intensive treatment unit are to maintain patient safety, and that includes not removing tubes. Keeping them (patients) so that they do not hurt themselves”. The effect that sedation can have on mental state can be prolonged, and restraints are often justified until the patient’s mental state can be assessed properly. Until then, life-saving therapies trump the reluctance which healthcare professionals may have to use restraints. The use of restraint is not favourable amongst healthcare staff for impairment management but a necessary evil for reducing falls and treatment interference [37,38]. Often, negative feelings are present towards the use of restraint, with a struggle between the duty of care and patient autonomy, which could sometimes foster feelings of guilt amongst healthcare staff [21]. Again, there is a difference in attitudes in other institutions. In a study in China, most healthcare professionals participating believed that restraint should be the first response when patients are at risk of endangering themselves [39]. Likewise, staff in that study often did not feel guilty when restraining patients [39]. However, in another study in which nurses based in a Chinese hospital were interviewed, the opposite was believed; restraint was often opposed due to its effect on patient dignity, and it was only used when relative consent was given [40]. This therefore shows that multiple sources need to be taken into consideration before casting judgements on the actions of healthcare staff in different environments.

## 10. Staffing Levels and the Use of Restraints

A non-patient-centred element which can influence the use of restraint is staffing levels. The fact that many restraints are applied outside of normal working hours would support this [41]. In some studies, a significant reduction in restraint was found when the register nurse/patient ratio was higher [31]. However, rather than the issue being enough staff numbers, it has been suggested that the issue is not having enough trained staff. Better outcomes and fewer restraints were found to be used when there were higher numbers of well-trained staff [42]. This can vary from setting to setting. In an ITU setting, having more than one patient per member of staff makes it more difficult to prevent patients from pulling out medical devices. Therefore, the use of restraints would be implemented due to the inability to be vigilant with multiple patients [37]. However, in another study conducted in Germany, lower nurse staffing ratios were not related to higher frequencies of restraint use. This indicates the variable nature of this research and the potential need for a systematic review [26].

Another method to improve and reduce the amount of physical restraint used would be improving the education of healthcare staff. This would ensure correct knowledge and skills when using restraints, as well as alternatives which could be used beforehand, meaning that physical restraint would be a last resort [40]. Nursing staff are at the forefront of making decisions about restraint, and thoughtful decision-making is required to consider many factors. By providing training and education with clear guidelines, an appropriate, ethical decision can be made [43,44].

## 11. Alternatives to Restraint

Restraint is often used as a non-challenging approach to difficult scenarios. However, avoiding the use of restraint may result in a reduction in the length of hospital stays [45]. Finding an appropriate non-resistant approach to care can be difficult but is optimal for patient autonomy and the prevention of the harmful consequences stated above. A clinical decision analysis model can be useful for determining a plan of action [46]. By looking at the reasons why restraints are used (for example, to reduce patient falls) and then assessing the situation and rectifying potential causes and hazards, restraint incidence can be reduced. Better lighting, lower beds, fewer obstacles and gym mats for a softer landing are examples which have been implemented with this goal in mind [47,48]. Other alternative practices involve patient–staff interaction and distraction. By listening to the wishes and needs of older patients and spending time with them, as well as encouraging the overall calm conduct of the nursing staff, patients can be settled as well as reducing the need for patients to inappropriately mobilise [18]. Also, encouraging them to spend time with other patients and arranging stimulating activities can improve the mental and social well-being of the patient and will keep them preoccupied. Supporting the patient’s mobility and the use of mobility aids can also give the patient back their autonomy and reduce fall risks without having to use restraint [18]. Therefore, by assessing the situation before applying restraints, solutions can be applied to the problems directly rather than brutishly using restraint as an easier option (Table 4). However, it is accepted that this solution is not always applicable in every situation, and more alternatives are needed [49]. Listening and verbal interventions, despite being frequently used, are often perceived to not be effective [50], but by means of better education, this perception could be changed.

## 12. Conclusions

The use of restraints in a non-psychiatric healthcare setting can involve some blurred lines due to the use of direct and indirect restraints. It was found that the justification used was to prevent the incidence of patient falls, but restraints were also implemented to ease pressures brought about by staff shortages. The finding that the use of restraint decreased when there was a better nursing skill mix involved in patient care was particularly interesting. It is consistently found that having more educated and experienced nursing staff leads to better patient outcomes [51,52]. Therefore, by increasing staff knowledge of the subject matter, less restraint could be implemented in reducing falls and pressures on decreased staffing levels. Techniques such as having open lines of communication between staff and patients can avoid the need for mobilisation and the incidence of falls. Admittedly, this adds more roles to an already stretched workforce; however, these techniques when implemented will still reduce the use of restraint and reduce the length of hospital stays [45]. 

To increase education with regards to how restraint is used in healthcare, both consciously and subconsciously, as well as techniques to avoid this, online learning or training sessions could be implemented at NHS trusts. Such programmes could present this information, showcasing that restraint is not necessarily the best solution for the patient as well as staff, whilst providing alternative techniques to deal with situations where it should be used. However, it is still widely noted that non-restraint techniques are preferable. Whilst improving communication and techniques to prevent or mitigate falls such as gym mats and mobility aids are viable techniques, a larger variety would decrease restraint use even further.

Assessing the attitudes towards restraint, it is often deemed a necessary evil. Staff dislike its use, but it is implemented for the good of the patient to prevent incidents such as falls or medical device interference. It is commonly accepted that the use of bed rails, tables and healthcare assistant specials are restraining, but it would be of interest to determine how many healthcare professionals consider it as restraint. Often, when considering restraint, the mind goes towards restraint devices such as wrist restraints which are now only used in exceptional circumstances. A survey to determine attitudes towards direct and indirect restraint and the use of non-restraint techniques would be very interesting and could possibly lead to the creation of a training session to improve awareness and knowledge of the use of restraint and the means of decreasing its prevalence.

## Figures and Tables

**Figure 1 geriatrics-04-00050-f001:**
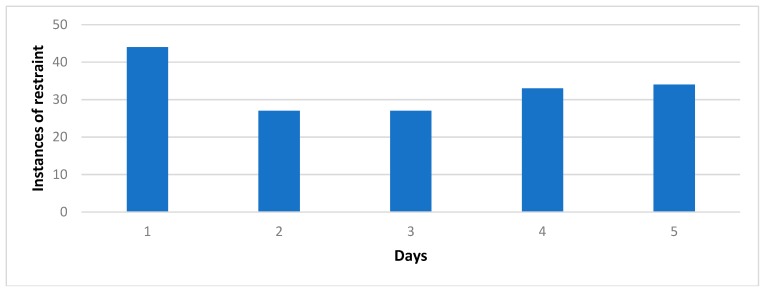
Total instances of restraint over five separate days on Acute Medical and Frailty Wards.

**Figure 2 geriatrics-04-00050-f002:**
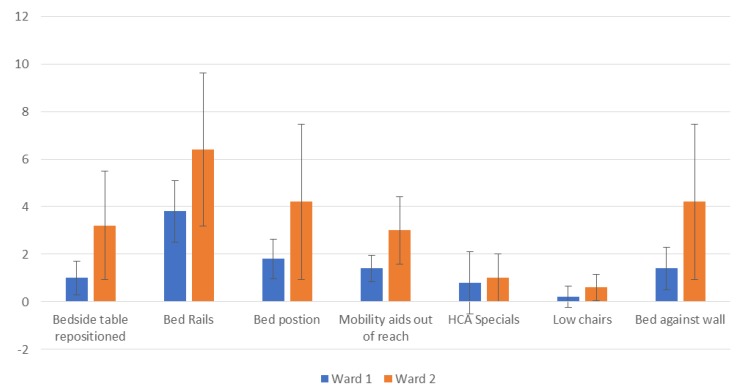
Average number of restraint instances per ward. Error bars represent standard deviation.

**Table 1 geriatrics-04-00050-t001:** Examples of direct and indirect restraint used in clinical practice.

Direct Restraint	Indirect Restraint
Movable tables	Passive interactions Out-of-reach mobility aids Persistent monitoring [sitters or specials] Lack of response to call bells
Low chairs	
Bed rails	
Restraint belts	
Bed linen [patients tucked into the bed]	
Locked doors	
Awkward bed positioning	
Bed positioned against a wall	
Mittens	
Web spacers	
Nasogastric tubes fixed in position	

**Table 2 geriatrics-04-00050-t002:** Consequences of restraint.

Physical	Urinary and Faecal	Mental
Skin Trauma	Constipation Urinary and faecal incontinence	Delirium
Muscle atrophy		Agitation
Limb injury, including fracture		Apathy
Skull fracture		Depression
Intracranial haemorrhage		Anxiety
Nerve injury [radial nerve/brachial plexus]		Aggression
Contractures		Frustration
Strangulation		Disempowered
Asphyxiation		Cognitive decline
		PTSD

**Table 3 geriatrics-04-00050-t003:** Example reasons for the use of restraints.

Staff	Device	Environment
Cultural	Maintaining a device in situ	Workload
		Lack of staff
Attitude		Ward layout
Coping strategies		Staff management
Defensive		Litigation
Role perception		

**Table 4 geriatrics-04-00050-t004:** Example reasons for disruptive actions by patients.

Examples that May Explain Behaviour
Longstanding routines [going to the kitchen at 6 pm to cook dinner]
Previous experiences [mine clearance or previous traumatic stress]
Pain [pain that they cannot verbalise]
Time frame [what year does the person with dementia think it is?]
Paranoia [thinking that people are after them, made worse by 1–1 monitoring]
Fear [strange surroundings, strange people]
Hallucinations and visions [caused by medication, infection or post-operative delirium]

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
