# Peer review of "The Attitudes Towards the Use of Restraint and Restrictive Intervention Amongst Healthcare Staff on Acute Medical and Frailty Wards—A Brief Literature Review"

_geriatrics, 2019, doi:10.3390/geriatrics4030050_

Round 1
Reviewer 1 Report
This is a review of an interesting topic. I think the title and abstract should make it clearer that the field of mental health is excluded. Another thing that detracts its value is being so circumscribed to the British sphere in some respects, while a literature review that includes other territories is then carried out. Although the format sometimes seems to be similar to a systematically review, works of very different quality are used. However, the quality of the works is not evaluated, because, after all, it is not a systematic review. That makes this manuscript have a very relative value. For example, authors talk about prevalence in the US through a reference of an opinion paper published almost 30 years ago. I think this paper lacks a lot of work to be close to deserving publication.Author Response
We thank the author for his comments.
We accept that the paper has a UK slant, but we have tried to include information from other geographical areas where possible.
We have removed the article Reference 13 and replaced it with a more up to date reference.
We have corrected syntax and grammatical errors where identified.
Reviewer 2 Report
first of all, I would like to congratulate the first author, a medical student, undertaking this review. The review, although rather comprehensive, contains also work that I assume was done as an audit, and it sits rather clumsy within the text - it is unclear where this was done, The reader is effaced with results form wards 1 and 2 etc, not knowing which hospital, which works this belongs to, whether the findings are published or not.
Although the MCA has been mentioned, there is no mention about the NICE stand on restrain, especially among people with dementia and delirium. this needs to be added and updated.
I would have liked a chart to inform the reader how many papers have been retrieved, how many papers were or interest/relevant to this review, and how many have been assessed. despite being a narrative review, this information would be beneficial to add.
Also, there is a need for the text to be reviewed, and some sentences attended to, in particular in terms of grammar and syntax.
Author Response
We would like to thank the reviewer for his comments.
We have added the NICE reference to the paper.
We have gone through the paper to correct errors as high lighted.
We have not included a chart regarding the papers but have added a section regarding the number of papers retrieved and reviewed.
Round 2
Reviewer 1 Report
The authors have corrected specific issues mentioned by the two reviewers but have not performed an extensive review of the manuscript. There are several problems. First, this article mixes a systematic review left halfway (which does not follow the PRISMA statement) with a small study carried out in a specific center. If they want to publish the review, they should finish it considering the PRISMA statement protocol. Reviews that do not follow this protocol should not be published. If they also want to add their study (which I am not quite sure, but I let the editor decide) they should perform calculations such as chi-square calculations of expected proportions to see if international figures fit your data, not just telling the story in a similar order.
Author Response
We would like to that the Reviewer for their comments and suggestions.
We have reviewed and amended the paper where we have identified issues with grammar and spelling.
This paper is a narrative review of the present situation in clinical care. We added the results of a small local audit, undertaken by the authors on the Acute Medical and Frailty units (ward 1 and ward 2 respectively). We believe this adds to the paper and provides additional support for the review.
As the paper is a narrative review and not a systematic review, we do not believe there is any need for us t adhere to guidelines surrounding systematic reviews. We agree with the Reviewer that should this have been a Systematic review, we would have need to following accepted guidelines.
Regards